# Multinomial Convolutions for Joint Modeling of Regulatory Motifs and Sequence Activity Readouts

**DOI:** 10.3390/genes13091614

**Published:** 2022-09-08

**Authors:** Minjun Park, Salvi Singh, Samin Rahman Khan, Mohammed Abid Abrar, Francisco Grisanti, M. Sohel Rahman, Md. Abul Hassan Samee

**Affiliations:** 1Department of Integrative Physiology, Baylor College of Medicine, Houston, TX 77030, USA; 2Computer Science and Engineering, Bangladesh University of Engineering and Technology, Dhaka 1000, Bangladesh; 3Computer Science and Engineering, Brac University, Dhaka 1212, Bangladesh

**Keywords:** convolutional neural networks, motifs, multinomial convolutions, multinomial convolutional neural networks, MPRA

## Abstract

A common goal in the convolutional neural network (CNN) modeling of genomic data is to discover specific sequence motifs. Post hoc analysis methods aid in this task but are dependent on parameters whose optimal values are unclear and applying the discovered motifs to new genomic data is not straightforward. As an alternative, we propose to learn convolutions as multinomial distributions, thus streamlining interpretable motif discovery with CNN model fitting. We developed MuSeAM (Multinomial CNNs for Sequence Activity Modeling) by implementing multinomial convolutions in a CNN model. Through benchmarking, we demonstrate the efficacy of MuSeAM in accurately modeling genomic data while fitting multinomial convolutions that recapitulate known transcription factor motifs.

## 1. Introduction

Convolutional neural networks (CNNs) have shown considerable accuracy in genomic data modeling, especially in the supervised modeling of DNA sequence readouts [1,2]. However, the models generally suffer from a lack of interpretability, necessitating post hoc analyses to interpret the trained CNNs [3,4,5,6,7]. Conceptually, a post hoc interpretation method first identifies segments in the input sequences where the model has discovered informative signals. Then, it consolidates those segments into a new model that summarizes the informative signals. Although post hoc approaches are informative, an important question is whether we could streamline the interpretation of CNN models through their training, particularly by making the convolutions directly interpretable instead of applying a separate post hoc pipeline. Such a streamlined approach could avoid the challenges associated with post hoc model interpretation, e.g., tuning the additional parameters and statistical thresholds of those methods, the optimal values of which are not always obvious. Furthermore, compared with the interpretations learned from post hoc approaches, convolutions that are interpretable could be more readily applied to new contexts.

To address the above question, we propose using *multinomial convolutions*—namely convolution matrices that represent multinomial distributions over the four nucleotides. The multinomial convolutions are directly interpretable as motifs for the DNA-binding specificity of transcription factors (TFs). Such motif models have been classically used in the biophysical modeling of genomic sequences. Thus, CNNs with multinomial convolutions do not require post hoc analyses for motif discovery and interpretation. Furthermore, similar to the classical motifs, the multinomial convolutions can be used in any new context—such as for scanning the genome and building alternative biophysical models of enhancer activity—independent of the original CNN model.

We implemented multinomial convolutions in a CNN model, MuSeAM (Multinomial CNNs for Sequence Activity Modeling). In regression modeling of enhancers in an MPRA (massively parallel reporter assay) dataset, MuSeAM showed high performance while also learning multinomial convolutions that recapitulate motifs of regulatory TFs in the corresponding cell line. However, this approach of “embedding” interpretability within the design may not result in a model that consistently outperforms alternative modeling choices requiring post hoc analyses [8]. Therefore, we feature multinomial convolution as a promising architectural choice, like dropout and batch normalization layers, when building CNN models for genomic data (also see Discussion).

## 2. Results

### 2.1. Multinomial Convolutions

To ensure that CNN convolution matrices are directly interpretable as classical sequence motifs, we learn them as multinomial distributions over the four DNA nucleotides. In particular, given a convolution matrix X, we transform X into a kernel T such that:$$T_{i,j}=\frac{e^{\alpha \cdot X_{i,j}}}{Z_{i}}$$where: $$Z_{i}=\overset{3}{\underset{j=0}{\sum }}e^{\alpha \cdot X_{i,j}}$$ and α is a free parameter of the model (Figure 1A, Methods). Here, i indicates the rows, and j indicates the columns of X (and T) in zero-based indexing. Thus, each row of T satisfies the conditions of a multinomial distribution (Methods), and we call T a multinomial convolution. Notably, the multinomial convolutions are directly interpretable as the widely used motif models of TF-DNA binding specificity. In the remainder of the paper, we use the term *multinomial motifs* to denote the multinomial convolutions of a trained CNN model.

Given a multinomial convolution matrix T of size L×4 and a “one-hot encoded” (Methods) nucleotide sequence S, we design the convolution operation to compute the following value at each L-length subsequence s of S:$$\sum_{i=0}^{L-1}\sum_{j=0}^{3}\ln\left(\frac{T_{i,j}}{B_{j}}\right)\cdot s_{i,j},$$

Here, B is a background distribution over the four DNA nucleotides. We call this value the LLR (logarithm of likelihood ratio) term for s because it denotes the logarithm of the ratio of likelihoods for generating s from T compared with generating it from B. In our study, we used the following non-uniform background distribution for human genomic sequences (A: 0.295; C: 0.205; G: 0.205; T: 0.295) [9].

The above approach offers several advantages compared with state-of-the-art approaches for deriving motifs through a post hoc analysis of deep neural networks. First, by directly modeling a multinomial distribution over the four nucleotides, the multinomial motifs become immediately interpretable as the classical motif models of DNA-binding proteins. Second, state-of-the-art post hoc analysis approaches derive motif models by clustering those segments in the input that the model considers informative [5,10]. Motif computation in these approaches is dependent on several statistical cut-offs, and the optimal selection of these values is not straightforward [5]. Multinomial convolutions are free from such issues. Third, the multinomial motifs can be readily used in new contexts, e.g., to perform whole-genome scans. This contrasts considerably with some of the most successful post hoc approaches that require first applying a trained neural network model to the new sequences, deriving the importance scores, and then applying the motif models to these values [7]. Finally, the multinomial convolutions are derived through background correction, a necessary step for building motifs from CNN models [5]. This enforces learning of specific signals while simultaneously optimizing the loss function. In particular, the gradient of the convolution operation output with respect to its input, i.e., $\frac{d\left(\ln\left(T_{i,j}\right)\right)}{d\left(X_{i,j}\right)}$, is reduced to $1-T_{i,j}$. Thus, minimizing the loss function becomes equivalent to maximizing the $T_{i,j}$ terms (i.e., probabilities in the multinomial distributions), which promotes learning of specific motifs. On the other hand, normalization by the background avoids learning $T_{i,j}$ terms that resemble the background distribution, i.e., degenerate motifs.

### 2.2. MuSeAM: A CNN Implementing Multinomial Convolutions

To assess whether a multinomial-convolution-based neural network can learn readily interpretable and biologically relevant motifs without sacrificing accuracy, we developed MuSeAM, a CNN model in which first-layer convolutions are multinomial. We computed likelihood ratios (LLR terms) for the multinomial convolutions and used these LLR terms as inputs to the successive layers in MuSeAM. Furthermore, we learned the architecture of these successive layers (such as the number of convolutional layers, the number of fully connected layers, and the activation functions) using 10-fold cross-validation (Figure 1B). Thus, except for the first convolutional layer, the architectural choices for MuSeAM remained the same as for any other CNN model.

Below, we apply MuSeAM in regression modeling of human enhancer MPRA (massively parallel reporter assay) data and benchmark it against alternative models based on conventional convolutions. We found MuSeAM learned biologically relevant motifs and demonstrated higher performance than the alternative models. We discuss MuSeAM’s findings regarding TFs corresponding to those motifs and present several novel motifs discovered by MuSeAM. Notably, the optimal architecture for MuSeAM in this dataset allowed us to easily infer each TF’s role (activator or repressor), which we discuss below in more detail.

### 2.3. Application of MuSeAM on Human Liver Enhancer MPRA Data

Massively parallel reporter assay (MPRA) is a high-throughput technology to quantify the enhancer activities of thousands of candidate sequences. Briefly, a candidate sequence is first placed in the context of a reporter gene in a synthetic construct. The sequence’s enhancer activity is then quantified as the normalized transcript count of the reporter gene in that construct [11]. For robust estimation, one can create many constructs for a given sequence; moreover, each construct contains a unique barcode, enabling a large set of independent observations of the sequence’s enhancer activity.

In our application of MuSeAM, we trained MuSeAM on a recent MPRA dataset of 2440 human DNA sequences assayed in the liver cell line HepG2 [12]. Inoue et al. synthesized each sequence with 100 unique barcodes, making this one of the most robust datasets available. We applied MuSeAM on this dataset under a 10-fold cross-validation scheme, using the mean Spearman correlation over the 10 folds as the model performance metric and optimizing the mean squared error. MuSeAM fit this dataset with a mean Spearman correlation coefficient of 0.52 (mean over 10 folds) (Methods). On the same dataset, the best-performing conventional CNN architecture achieved a mean Spearman correlation value of 0.42, establishing that the multinomial convolution approach of MuSeAM demonstrates considerably higher performance for this dataset. We determined the optimal architectures of both MuSeAM and the conventional CNN models through an extensive hyperparameter search. We provide the details in the Methods section; briefly, the optimal architecture for MuSeAM had a single convolution layer with 512 filters—each of length 12 and with ReLU activation—followed by a maximum pooling, a dropout, and a fully connected layer. For the conventional CNN model, the optimal architecture had five convolution layers, where all filters were of length five and used ELU activation, followed by a max pooling layer, a dropout layer, and a fully connected layer.

### 2.4. Multinomial Convolutions in MuSeAM Reveal Regulatory TFs in Human Liver

Next, we performed a series of analyses to determine whether MuSeAM successfully learned biologically relevant motifs. For any de novo motif discovery approach, it is essential that the learned motifs are specific (not degenerate). We generally analyze the learned motifs in terms of their overall similarity with existing motif models and identified the TFs relevant to the dataset. Such analyses become reliable if the de novo motifs are specific. Motif specificity is also necessary for their meaningful application beyond the learned model, such as in genome-wide scans or in more biophysically grounded models of enhancer activity. Thus, we first checked the specificity of the multinomial convolution motifs. We compared the information content per position (ICP) of the multinomial convolution motifs against the ICP of known motifs from public databases [13,14] (Figure 2A). We used a Welch two-sample *t*-test, which rejected the hypothesis that database motifs are more specific than MuSeAM motifs (*p*-value $<2.2\times 10^{-16}$; Methods).

Next, we checked if MuSeAM motifs correspond to TFs with known regulatory functions in the human liver. It is a fundamental expectation that an MPRA model will identify the TFs that regulate a reporter gene’s expression [15]. Such a model should also infer each TF’s role, i.e., whether it is an activator or a repressor of gene expression. It is straightforward to find both sets of information in MuSeAM. First, we identify the relevant TF by searching motif databases [13,14] for matches to each MuSeAM motif. Second, there is a one-to-one correspondence between the multinomial convolutions and the fully connected layer weights because the architecture is a linear regression model of the max-pooled LLR terms. Thus, the sign of the fully connected layer weights denotes whether the corresponding TF is a potential activator (positive weight) or repressor (negative weight). Importantly, because MuSeAM fits the multinomial convolutions in a data-driven manner, a database search may not necessarily find a hit for every multinomial convolution. These examples would be indicative of novel motifs.

We found matching TFs for 506 motifs using a motif database search for the multinomial convolution motifs. We further confirmed these 506 TFs are expressed in the HepG2 cell line (using ENCODE RNA-seq data; see Methods). Our analysis therefore shows that MuSeAM can identify known activating and repressing TFs. For example, the multinomial convolutions with the highest positive weights in the fully connected layer matched with FOSL2, FOS, BATF, HNF1A, and HNF1B (Figure 2B). All these TFs are known to function as activators [16,17,18,19,20]. Likewise, the multinomial convolutions with the most negative weights in the fully connected layer matched with known repressive TFs, including SMAD2 and TGIF-1, a transcriptional co-repressor of SMAD2, FOXP3, HES1, and ZBTB6 (Figure 2C) [21,22,23,24,25,26]. Our analysis also revealed six novel motifs—one activator and five repressors (Appendix A). However, although these motifs did not have matches in the motif database, they are not degenerate—their specificities are comparable to known motifs (Figure 2D,E).

## 3. Discussion

The use of CNNs is now routine in modeling genomic datasets [1,2,27,28]; however, CNN model interpretability remains a concern [10]. Post hoc analysis approaches have been useful in eliciting sequence motifs from a trained CNN model [5]. However, for the reasons mentioned above, we proposed implementing multinomial convolutions. The multinomial convolution formulation is particularly attractive because of its immediate interpretability as the widely used and biophysically understandable sequence motif. Furthermore, it does not make any assumptions about the concentration of TFs [28] or require any post hoc analysis [10]. Finally, multinomial convolution motifs are interoperable and can be used with a large array of genomic data analysis pipelines that are currently being utilized for applications such as sequence annotation and motif enrichment. Therefore, we anticipate that multinomial convolutions will find broader use in the future. We also envision that multinomial convolutions will enable an increased use of more biophysically grounded functions [29] in the intermediate layers of CNN models of genomic data.

In our comparative analysis, our implementation of multinomial CNNs, namely MuSeAM, consistently provided better results than those derived from the alternative architectures of conventional CNN models, implying that multinomial convolutions can serve the intended purpose of learning directly interpretable motifs without sacrificing model accuracy. However, this approach to designing readily interpretable models may not always outperform conventional CNNs. Thus, we propose that multinomial convolutions be treated as an architectural choice during CNN model architecture searching, similar to dropout layers and batch normalization. We believe that evaluating them side-by-side with the conventional convolutions through an architecture search scheme, such as cross-validation, will prove valuable for CNN modeling of genomic data.

To our best knowledge, the notion of multinomial convolutions has not previously been used for the purpose of motif learning in CNN models. The idea of multinomial convolutions is akin to “soft-max” normalization; however, we are not aware of any studies mentioning soft-max-normalized convolutions that demonstrate their utility similarly to our research, i.e., using the distributions to generate a biophysically interpretable and likelihood-based score for the input to a deep neural network. A preprint by Ploenzke and Irizarry [30] used a different approach to achieve the same goal. However, the use of multinomial distribution in our approach ensure it is efficiently trainable in any CNN optimization scheme. Moreover, the approach of Ploenzke and Irizarry requires constrained parameter optimization and readjustment, and its efficacy and computational efficiency have not been demonstrated in real large-scale genomic datasets such as in the case of our method in this study. Additionally, Koo and Ploenzke showed that exponential activation in shallow networks [31] can facilitate post hoc derivation of interpretable motifs. However, multinomial convolutions do not require post hoc analyses, and a multinomial CNN model does not need networks to be shallow.

## 4. Methods

### 4.1. MPRA Data

Data were from the liver enhancer dataset of Inoue et al. [12].

### 4.2. Motif Databases and Motif Searching

We used motifs from the HOmo sapiens COmprehensive MOdel COllection (HOCOMOCO)-v11 database of transcription-factor-binding models to compare the multinomial convolution motifs against currently known motifs [13]. The current HOCOMOCO release includes 769 motifs that represent the DNA-binding specificity of 680 TFs. We used the Tomtom tool from MEME Suite for the motif search [32].

### 4.3. RNA-seq Data

Processed RNA-seq data of the ENCODE consortium [33] were collected from https://genome.ucsc.edu/ENCODE/downloads.html; accessed on 1 August 2021.

### 4.4. MuSeAM: A Multinomial Convolutional Neural Network Model

MuSeAM is a convolutional neural network model with a custom-implemented convolutional layer. The remaining MuSeAM architectural choices were decided based on cross-validation, as is commonly conducted for CNN models. We implemented MuSeAM using Keras in TensorFlow [34].

Input Layer

For each sequence, the model receives two sets of one-hot encoded inputs: one is a representation of the forward strand of the sequence, and the other is the reverse complement of the same. The one-hot encoding of a genomic sequence S of length N is an N×4 matrix $\mathrm{Enc}\left(S\right)$ where the i-th row is a vector encoding the identity of the i-th nucleotide of S as follows.$$\mathrm{Enc}{\left(S\right)}_{i}=\left\{\begin{array}{c} \left(1,0,0,0\right),\;\;\mathrm{if}\;S_{i}=A\\ \left(0,1,0,0\right),\;\;\mathrm{if}\;S_{i}=C\\ \left(0,0,1,0\right),\;\;\mathrm{if}\;S_{i}=G\\ \left(0,0,0,1\right),\;\;\mathrm{if}\;S_{i}=T \end{array}\right.$$

For ease of notation, S denotes below both the DNA sequence and its one-hot encoding.

Convolution Layer

The convolution layer in MuSeAM is a custom implementation where each convolution learns multinomial distributions as detailed in Results.

Convolution and Pooling operations

For each multinomial convolution matrix T of size L×4 and one-hot encoded nucleotide sequence S of length N, a convolution operation computes the following term on each L-length subsequence s of S.

$$\sum_{i=0}^{L-1}\sum_{j=0}^{3}\ln\left(\frac{T_{i,j}}{B_{j}}\right)\cdot s_{i,j}$$ where B is a background distribution over the four DNA nucleotides. In other words, for each L-length subsequence s of S, the convolution operation computes the logarithm of the ratio of two likelihood terms. The first (the numerator) is the likelihood of generating the sequence s from the convolution matrix T (i.e., its multinomial distribution). The second is the likelihood of generating s from the background. In our current analysis, we used the following distribution as our background distribution B: (A: 0.295; C: 0.205; G: 0.205; T: 0.295).

The convolution operation produces a vector of N−L+1 log-likelihood ratio (LLR) terms for each of the two incoming input units (corresponding to the forward strand and reverse-complement representations of the input sequence S). We passed the LLR terms through activation and pooling functions.

Output Layer

The output from the convolution layer is passed on to a fully connected layer, which yields the final output value.

### 4.5. Optimization and Model Selection

Loss Function

MuSeAM optimizes the parameters using the mean squared error (MSE) function as the loss function. Given a batch of n sequences with the true and predicted values of the i-th sequences being $y_{i}$ and $\hat{y_{i}}$, respectively, the MSE function computes the following term:$$MSE=\frac{1}{n}\sum_{i=0}^{n-1}{\left(y_{i}-\hat{y_{i}}\right)}^{2}$$

Model Optimization

We compiled MuSeAM to optimize MSE loss with the *adam* optimizer and reported the model performance in terms of Spearman rank-order correlation. We randomly initialized all weights and bias terms in the model using the *glorot uniform* initializer.

Model Selection

We used a 10-fold cross-validation to select the optimal values for the following parameters: number of convolutions, size of each convolution, regularization type, number of epochs, and batch size. For each fold, we partitioned the data into 10% test, 9% validation, and 81% training sets. We recorded the Spearman correlation coefficient for each fold and used the mean over the 10 folds to determine the best set of learning parameters.

### 4.6. Comparing Information Content per Position

Given a motif M of length L, we first normalized the elements in each row of M by the sum of the row. Then, we calculated the information content per position (ICP) of the normalized motif as:$$\frac{1}{L}\sum_{i=0}^{L-1}\sum_{j=0}^{3}M_{i,j}\cdot \log_{2}\left(\frac{M_{i,j}}{B_{j}}\right)$$

Here, B represents the background distribution. Given the ICP values of the HOCOMOCO motifs [13] and the multinomial convolution motifs, we performed a Welch two-sample *t*-test using the R command *t*-test (data1, data2, alternative = “less”, var.equal = False), where data1 and data2 are the ICP values of the HOCOMOCO and multinomial motifs, respectively.

### 4.7. Alternative Architectures for MuSeAM and Conventional CNN

We tested various alternative architectural choices for both MuSeAM and the conventional CNN model to obtain the model architecture that gave us the highest mean Spearman correlation under 10-fold cross-validation. The hyperparameters selected for the final architecture for both models were those that resulted in the highest performance.

MuSeAM

The tested hyperparameters and their range of variation are as follows:Number of filters from 64 to 1024;Filter lengths from 5 to 16;One and two fully connected layers;α values from 100 to 140;Batch sizes from 32 to 512;Regularization factors from 0.0005 to 0.01;Dropout rates from 0.001 to 0.6;Activation functions from ReLU, sigmoid, and tanh;Pooling approaches from max, sum, and average;Early stopping.

Conventional CNN

The tested hyperparameters and their range of variation are as follows:Convolution layers from 1 to 5;Number of filters from 64 to 1024;Filter lengths from 5 to 16;One and two fully connected layers;Batch Sizes from 32 to 512;Regularization factors from 0.0005 to 0.01;Dropout rates from 0.1 to 0.6;Activation functions from ReLU, ELU, sigmoid, and tanh;Pooling approaches from max, sum, and average;Early stopping.

### 4.8. Optimal Hyperparameters

The final MuSeAM model has one convolution layer with ReLU activation, followed by max pooling, dropout, and fully connected layer. We reached the highest mean Spearman correlation using 512 convolutions, each of a length of 12, and with a single node in the fully connected layer. The dropout rate was 0.4. We divided the data into batch sizes of 64 and used early stopping while training. We used a learning rate of 0.001 and a regularization factor of 0.001.

We achieved peak conventional CNN performance using five convolutions layers, ELU activation, a max pooling layer, a dropout layer, and a single-node fully connected layer. The number of convolutions for each of the five layers was 512, 256, 64, 64, and 64, respectively, and the filter length for each was 5. The optimal dropout rate was found to be 0.5, and the data were partitioned into batch sizes of 32. We applied early stopping.

## Figures and Tables

**Figure 1 genes-13-01614-f001:**
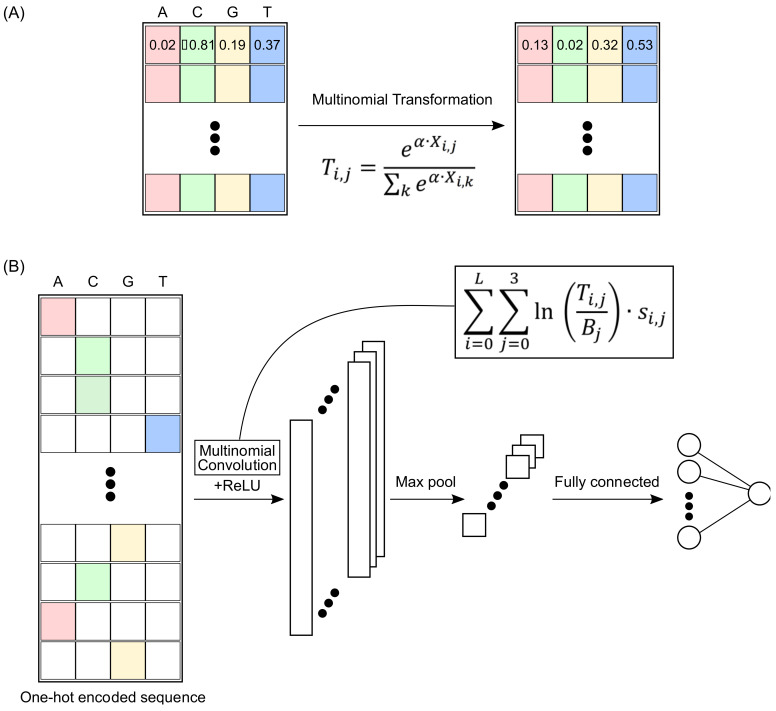
(**A**) Mathematical representation of multinomial transformation of convolutions. (**B**) Overview of multinomial CNN (MuSeAM) model with custom convolution operation.

**Figure 2 genes-13-01614-f002:**
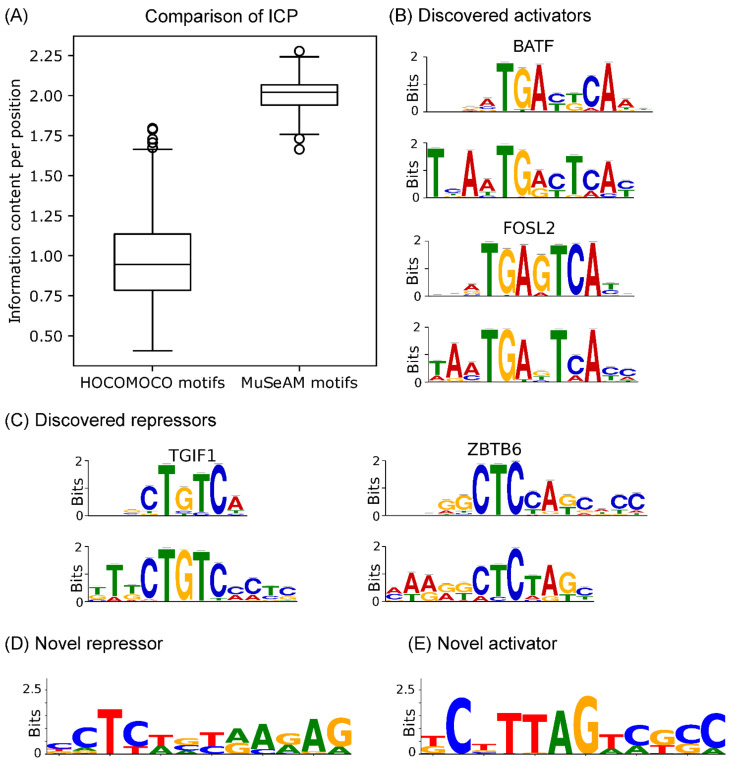
Modeling of human liver enhancer MPRA data: (**A**) information content per position for known motifs (**left**) and MuSeAM motifs (**right**); (**B**) motif logos of best-match known motifs (**top**) and MuSeAM-discovered motifs (**bottom**) for activators (positive weight); (**C**) motif logos of best-match known motifs (**top**) and MuSeAM-discovered motifs (**bottom**) for repressors (negative weight); (**D**) novel repressors (negative weight); and (**E**) novel activators (positive weight) identified by the MuSeAM model.

## Data Availability

https://github.com/SameeLab-BCM/MuSeAM accessed on 1 July 2022.

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
