# Peer review of "Multinomial Convolutions for Joint Modeling of Regulatory Motifs and Sequence Activity Readouts"

_genes, 2022, doi:10.3390/genes13091614_

Round 1
Reviewer 1 Report
The Manuscript entitled: Multinomial Convolutions for Joint Modeling of Regulatory Motifs and Sequence Activity Readouts has been reviewed.
Major Queries/Comments:
· This introduction is not well written. Please, you must restructure it and write it again.
· Results must be improved. They are not clearly presented.
Minor Queries/ Comments
1. The Manuscript could benefit from editing grammar, missing words, subject-verb agreement, etc. It is recommended that authors delete irrelevant "general" phrases and sentences and repeated and unnecessary words. They should use short sentences. Also, some Introductory sentences are unrelated or are not needed. Authors should also send their Manuscript to an expert in English editing and academic writing.
2. The figures are not precise. Please modify the resolution and the size of the figures.
3. Page 6 … repetitive titles (Convolution Layer & Multinomial Convolution Matrices) … Please delete one.
4. Page 12 … Please correct the reference (2017b).
Best regards,
Reviewer 2 Report
In this article, the author mentioned that the notion of multinomial convolutions has not been used before for the purpose of motif learning in CNN models. The idea of multinomial convolutions is akin to “soft-max” normalization. However, there are not aware of any work mentioning soft-max normalized convolutions and demonstrating their utility in the way we did, i.e., using the distributions to generate a biophysically interpretable and likeli-hood-based score that goes as the input to a deep neural network. Therefore, in this paper, the author proposes a new approach .The propose to learn convolutions as multinomial distributions, thus streamlining interpretable motif discovery with CNN model fitting and implemented Mu-SeAM (Multinomial CNNs for Sequence Activity Modeling), a CNN model using multinomial con-volutions. The benchmarks in this article show the efficacy in accurately modeling genomic data while fitting multinomial convolutions that recapitulate known transcription factor motifs. The experimental design and data are clear, and the ideas proposed are novel. However, there are some problems in the details, mainly as follows:
(1).In Figure1, the font in the figure is small and not very clear. The same phenomenon also occurs in Figure2, and the picture in Figure2 is too large.
(2).The image and image name in Figure S1 are not on the same page. The title of the picture and its explanation should be on the same page. The font in the picture is too small to be clear.
(3).You should attach a number to the subheading of each chapter, which will make the article logical.
(4).The format of the article is not uniform, and there are many formulas mistakes in the article, some of the interval is relatively small, some of the interval is relatively large.
(5). It is noted that the manuscript needs careful editing by someone with expertise in technical English editing paying particular attention to English grammar, spelling, and sentence structure so that the goals and results of the study are clear to the reader.
Reviewer 3 Report
(1) The manuscript needs an English proofreading.
(2) What is (are) the limitation(s) of the proposed approach?
Round 2
Reviewer 1 Report
The authors have satisfactorily responded to all my questions. I have no additional comments. Thank you.